# MEMS Scanning Mirrors for Optical Coherence Tomography

**Christophe Gorecki** [1,*] **and Sylwester Bargiel** [2]

1. International Center for Translational Eye Research, Institute of Physical Chemistry, Polish Academy of Sciences, Skierniewicka 10A, 01-230 Warsaw, Poland
2. FEMTO-ST Institute (UMR CNRS 6714/UBFC), 15B Avenue des Montboucons, 25030 Besançon, France; sylwester.bargiel@femto-st.fr
* Correspondence: cgorecki@ichf.edu.pl

**Abstract:** This contribution presents an overview of advances in scanning micromirrors based on MEMS (Micro-electro-mechanical systems) technologies to achieve beam scanning for OCT (Optical Coherence Tomography). The use of MEMS scanners for miniaturized OCT probes requires appropriate optical architectures. Their design involves a suitable actuation mechanism and an adapted imaging scheme in terms of achievable scan range, scan speed, low power consumption, and acceptable size of the OCT probe. The electrostatic, electromagnetic, and electrothermal actuation techniques are discussed here as well as the requirements that drive the design and fabrication of functional OCT probes. Each actuation mechanism is illustrated by examples of miniature OCT probes demonstrating the effectiveness of in vivo bioimaging. Finally, the design issues are discussed to permit users to select an OCT scanner that is adapted to their specific imaging needs.

**Keywords:** micro-opto-electro-mechanical system; MEMS scanner; optical coherence tomography

## 1. Introduction

Micro-electro-mechanical systems (MEMS) technology enables the building of microoptical scanners that are well suited for low cost manufacturability and scalability as the MEMS processes emanate from the mature semiconductor microfabrication industry [1]. For a long time, the potential of MEMS to steer or direct light has been well demonstrated in the field of free-space optical systems [2]. In the 80s and early 90s, telecommunications became the market driver for the optical applications of MEMS, pushing the development of scanning micromirror systems for optical switches and network ports [3]. More recently, many types of MEMS scanning mirrors have been developed, covering a wide range of applications from micrometer-scale array-type components to large scanners for high-resolution imaging [4]. Thus, numerous optical imaging techniques such as confocal microscopy [5,6], multiphoton microscopy [7,8], and Optical Coherence Tomography (OCT) [9–11] have become important diagnostic tools in biomedicine, particularly offering a platform for endoscopic imaging. These MEMS scanners successfully replaced the bulky and high power consuming galvanometer scanners, providing compact, low cost, and low power consumption solutions for high speed beam steering. Further, 2D MEMS mirrors that scan in two axes are a pertinent alternative to the large galvano-scanners [12].

The MEMS scanner's performances are closely linked to the size of the selected actuator, carrying the micromirror and the force developed by this actuator. Figure 1 represents a summary of scanning micromirror applications, including the corresponding actuation mechanisms and main microfabrication technologies [13]. At the scale level, going from 1 mm to 1 cm, the MEMS technology combined with fiber optics enables miniature scanning components to be embedded inside the endoscopic imaging probes operating at high speed and high resonance frequency. The MEMS scanners are relatively easily integrated and adapted for low cost fabrication and low power consumption. The miniaturization performances and subsequent advances in standardized micromachining

technologies have also offered numerous low cost and disposable OCT probes for the medical industry. Originally adopted by the ophthalmic community [14–16], OCT has been used to image internal organs, such as the gastrointestinal tract [17], and in the diagnosis of skin pathologies [18,19]. This strong interest for clinical applications pushed several companies to develop endoscopic OCT systems [20]. Examples of commercial products are the clinical endoscope and catheter-based systems from the NvisionVLE® Imaging System (South Jordan, Utah, USA) [21], the intravascular OCT imaging systems from OPTIS™ (St. Jude Medical Inc., St. Paul, MN, USA) [22], Santec's (Komaki, Japan) swept-source OCT systems [23], Thorlabs (Newton, NJ, USA) OCT scanners [24], as well as Mirrorcle (Richmond, CA, USA) microscanners [25].

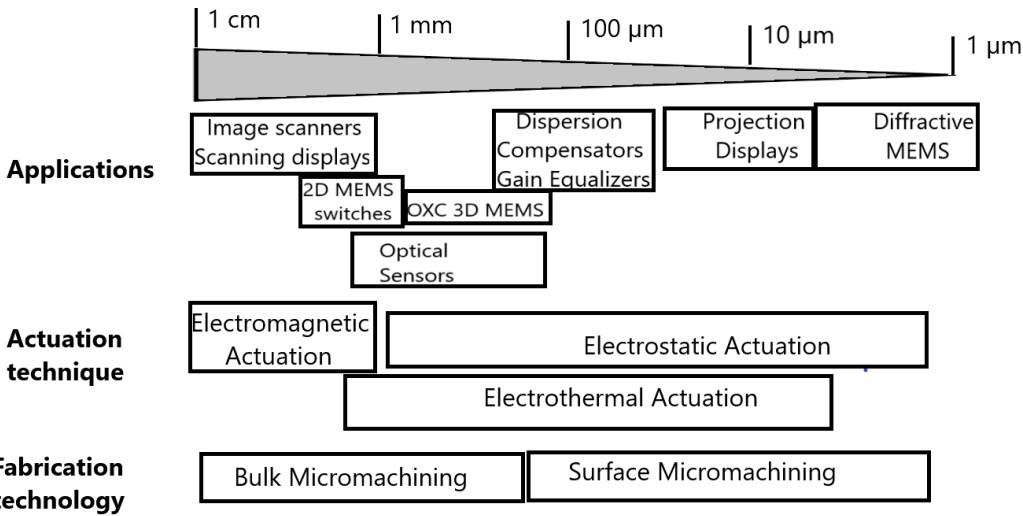

**Figure 1.** Applications, actuation mechanisms, and fabrication technologies for scanning micromirrors.

In this paper, we will demonstrate that for OCT imaging applications, the performance of the MEMS scanner is often limited by optics and intrinsic characteristics of actuation mechanisms. Here, optics require a small focused spot and dynamic focusing, imposing severe restrictions on scanning lens performances, while the actuation needs a high scanning speed, a low power consumption, a precise control of motion linearity, and reduced cross-axis coupling, which may distort the scanning patterns [26,27]. The group of OCT probes to be discussed in this paper do maintain such opto-mechanical performances, using different actuation mechanisms. Our wish is to demonstrate that the breakthrough of compactness is obtained when MEMS dual-axis beam-steering micromirrors [28] are used to achieve scanning 3D OCT probes. In the case of endoscopic applications, they are small enough to be included into a standard endoscope channel, with an inner diameter of 2.8 mm. Further, 2D scanning motion can derive from electrostatic, electromagnetic, electrothermal, or piezoelectric actuation, providing the scanning mirrors for light beam steering, operating at high speed and fully controlled by non-resonant or resonant regimes. However, we intentionally excluded from the present study piezoelectric actuation and we consider only the three other actuation mechanisms for MEMS scanning mirrors that are the most widely used in OCT applications.

## 2. Requirements for MEMS Microscanners

From the user point of view, performances of scanning micromirrors are defined by the maximum scan angle, the number of resolvable spots which represents the scan resolution, the resonance frequency, as well as the surface quality vs. the smoothness and flatness of micromirrors.

The number of resolvable spots $N$ of a scanning mirror is defined as a function of optical scan angle $\theta_{opt}$ and beam divergence $\delta\theta$, as shown in Figure 2:

$$N = \frac{\theta_{opt}}{\delta\theta} = \theta_{opt}\frac{D}{a\lambda},$$ (1)

where $D$ is the mirror diameter, $a$ represents the aperture shape factor ($a = 1$ in the case of a square aperture and 1.22 for a circular aperture) and $\lambda$ is the illumination wavelength.

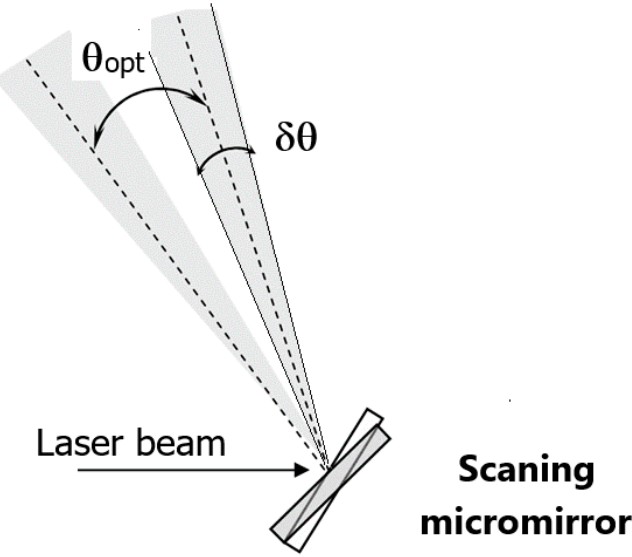

**Figure 2.** Total optical scan range of a scanning micromirror vs. the beam divergence.

As the number of resolvable spots is proportional to the product of mirror diameter by the scan angle, the resonant frequency defining the upper limit of the scanner's response depends on the diameter (or inertia) and mechanical rigidity of the mirror suspension. In the case of a circular mirror with a diameter of 1 mm, illuminated by a He-Ne laser and scanning in the mode of VGA ($640 \times 480$ pixels), $\theta_{opt} = 28.3°$, which is two times bigger than the mechanical deflection. In this case, the scanning mirror needs to deliver a full mechanical angle $\theta_{mech} = 14.15°$ or larger.

To make the comparison of MEMS scanners operating at different frequencies also, we use the $\theta_{opt} D = 1.22N\lambda$ product, which is the key performance index for scanners.

Imaging applications for 2D scanning require a dual axis system with fast scanning capability. Two categories of more popular MEMS torsional scanners are the uniaxial scanner and the biaxial gimbal-mounted scanner [29]. Figure 3 shows both these scanning architectures with associated scanning trajectories. Here, two full raster cycles are represented while the scanning spots are schematized in green. The uniaxial scanner has a single axis of rotation and to obtain a 2D scanning, a pair of uniaxial scanners is used (one horizontal, one vertical). Here, the rotation of the first scanner causes the optical beam to walk across the second scanner. In this case, the writing of data is obtained only during the forward sweep of the horizontal scanner. The biaxial scanner includes two perpendicular axes of rotation. The oscillation of the inner frame creates a horizontal scan line, while the outer frame creates the vertical scan line. This is writing a new line of data in both scan directions. Thus, the scanner writes two lines during one scan cycle.

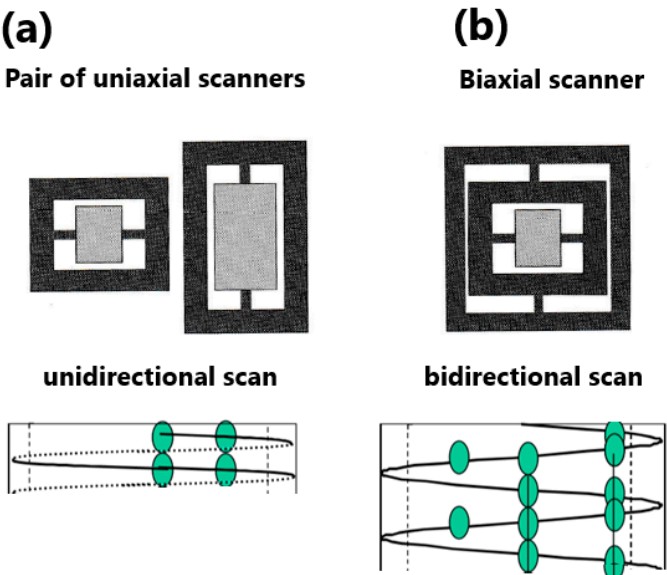

**Figure 3.** Schema of uniaxial scanner in (**a**) vs. biaxial scanner in (**b**) with associated scanning trajectories.

The performance of the MEMS optical scanner is often limited by the optical architecture, which requires small, focused spots and dynamic focusing. In addition, the intrinsic micromechanical characteristics of the actuation mechanism of the MEMS scanners are crucial and dependent upon the requirements of high scanning speed, low power consumption, as well as a precise control of motion linearity. Here, one of the limiting features in MEMS scanners is their mode of driving, which can be resonant or quasi-static. A resonant drive mode with a fine control of the beam's location is more difficult to implement, requiring one to use the close-loop control of beam steering. A quasi-static mode, on the other hand, provides easy programmable control of the beam, permitting operation with open-loop control of beam steering. However, certain categories of MEMS scanners are not able to offer the significant aperture size and scan angle specifications in the quasi-static mode. The quasi-statically tilted MEMS scanner only requires open-loop control for beam steering purposes. The main challenges of a bidirectional scan are the precise control of phase between both the scan lines and a high-quality control of motion, avoiding the effects of mechanical coupling between the scanner axes.

The choice of appropriate scanning techniques is strongly dependent on the selection of optimal scanning frequency. The most common scanning techniques are raster scanning and Lissajous scanning, shown in Figure 4 [30]. In raster scanning, a low frequency, linear vertical scan (usually in quasi-static mode) is paired with an orthogonal high frequency, resonant horizontal scan. For the raster scanner, the vertical scan is often assumed to be 60 Hz (video frame rate). The raster scanning can be obtained by bidirectional scanning waveforms with $N = f_x/f_y$, where $N$ is a positive integer. A raster scanner involves a triangular trajectory in the $x$-axis while shifting the sample position in steps or continuously in the $y$-axis. Here, a laser beam is starting at the top left of Figure 4a and goes from left-to-right at a scanning frequency of $f_x$ to the bottom-right corner (1), then rapidly moves back to the left and scans the next line (2), then off once again to go back up to the top (3). During the period of one scan cycle, the vertical position increases steadily downward at a frequency of $f_y$, which is the slow scanning frequency. The scan resolution and the frame rate are, respectively, defined by the lateral and vertical scanning frequencies. Raster scanning results in a rectangular scanning area. This simple technique of scan presents several technical limitations for MEMS micromirrors such as the higher driving voltages due to the use of non-resonant motion and lower mechanical stability for the slow axis. Figure 4b shows an alternative to raster scanning which is biresonant Lissajous scanning. This is achieved by driving the x and y axes with purely sinusoidal signals of

different frequencies $f_x$ and $f_y$ that are x(t) = $A_x$ cos($2\pi f_x$t) and y(t) = $A_y$ cos($2\pi f_y$t). The corresponding scanning waveforms have a frequency relation of $n = f_x/f_y$, where $n$ is a rational number. Lissajous scanning provides a rectangular scan area too. The scanning trajectory and the frame rate are more complex than in raster scanning trajectories and MEMS mirrors are operating at high resonant frequencies for both the axes, offering better mechanical stability.

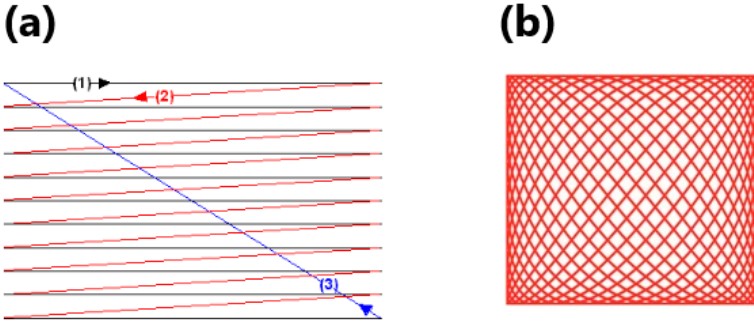

**Figure 4.** Principle of main scanning techniques: (**a**) raster scanning and (**b**) Lissajous scanning.

Figure 5a shows a schematic of a rectangular torsional scanning mirror with rectangular flexure beams [31]. Applied to produce the desired torsional mode, this torsional scanning mirror presents three parasitic oscillation modes that we refer to as vertical (or piston), horizontal, and rocking modes. In the ideal design, the mode frequencies should be well separated from the torsional mode frequency and its harmonics to minimize the power dissipation. Figure 5b shows that the horizontal and vertical modes do not change the scanned beam direction. Rocking mode deflects the incoming beam perpendicular to the intended scan axis, creating an undesired off-axis motion. The biaxial scanners should be designed in such a way that cross coupling effects between the torsional and rocking modes of inner and outer scan frames are minimized.

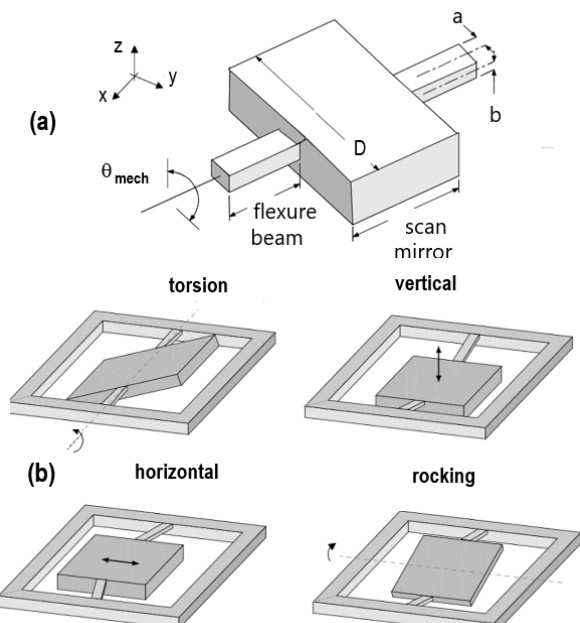

**Figure 5.** Torsional resonant scan mirror in (**a**) with its four fundamental vibration modes in (**b**).

Actuated MEMS mirrors are subjected to both static and dynamic deformations. Static deformation is induced by intrinsic material stress or thermal stress of mirror material,

while dynamic deformation is produced by forces due to mirror oscillation with a frequency *f*. The maximum mirror deviation from linearity according to Brosens's formula [32] is:

$$\delta_{max} \varpropto= \frac{\rho \; f^2 \; D^5 \; \theta_{mech}}{E \; t_m^2} \; , \tag{2}$$

where ρ is the material density, *E* the modulus of elasticity, and $t_m$ represents the mirror thickness.

For a fixed $\theta_{opt} \, D$ product, an increase of *D* leads to larger mirror deformation, lower maximum frequency, and increasing footprint. To keep the spot diffraction limited, the maximum mirror deformation should not exceed λ/10. Here, the $D^5$-dependency determines the upper limit of mirror size, while large mirrors must have a large thickness $t_m$. It can be roughly estimated that the optimum mirror surface will be around 1 mm².

## 3. Requirements for OCT Probes

A typical OCT setup includes a Michelson interferometer and a low-coherence light source. Interference signal, carrying the information about the measured biological object, is detected and demodulated to produce a map of the light backscattered from the microstructure inside the measured tissue. Image reconstruction is obtained by repeated axial measurements at different transverse positions as the optical beam is scanned by a MEMS scanning mirror. Figure 6 shows the two classes of miniature OCT probes, categorized on the basis of their scan modes: the side-imaging and the forward-imaging [33–35]. Side-imaging probes, schematized in Figure 6a, are the most widely used because they tend to have a much simpler actuation mechanism than forward-imaging probes and the actuator tends to be far away from the probe output. This category of OCT probes is very flexible and has a small size that is well appropriate for the miniaturization. Here, a mirror or prism is connected, for example, to a rotation assembly that shifts the emitted light from the optical fiber out of a window on the side of the probe. The side-imaging OCT probe only provides side imaging around the probe, which limits its clinical applications, making the image-based surgical needle guidance difficult. The forward-imaging probe, of which its size tends to be in a range of needle sizes, is more suitable for surgical guidance inside the body. However, forward-imaging probes are generally more complex in design. They require the actuator near the probe tips and are quite difficult to miniaturize. Here, a mirror or prism assembly shifts the emitted light from the optical fiber out of a window on the front side of the probe, as shown in Figure 6b.

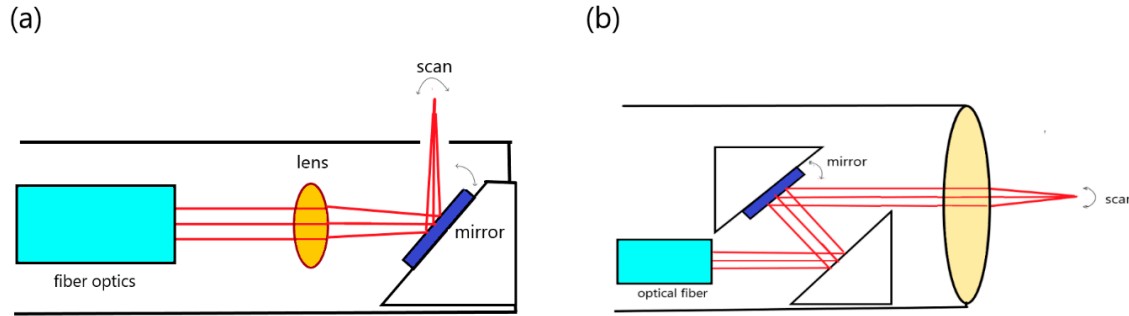

**Figure 6.** Two categories of miniature OCT probes: (**a**) side imaging probe; and (**b**) forward imaging probe.

In addition to the categorization of Figure 6, the scanning arrangement exhibits two distinct arrangements depending on the position of the MEMS scanner and the objective lens focusing the beam on the sample to be measured: pre-objective scanning and post-objective scanning [36,37]. In the pre-objective scanner, a MEMS mirror is placed prior to an objective lens, allowing a long working lens-sample distance. However, this configuration can cause significant off-axis aberration for a deflected scanning beam. The aberration can be minimized by using a specialized f-Theta lens, focusing the laser beam on a single

plane over the entire scan field while also ensuring that the beam remains perpendicular to the plane over the entire scan. In this category of probes, the miniaturization becomes difficult. In the post-objective scanner, a MEMS scanner is located after an objective lens. This configuration can produce a small off-axis aberration as well as high image resolution. However, the working distance becomes short and thus, the selection of an objective with a high numerical aperture objective is necessary. As the deflection by a mirror results in a curved focal plane, it is possible to compensate this curvature by varying the focal length of the lens by moving the lens along the beam path.

Finally, the last categorization of OCT probes refers to the location of the scanning mechanism. The scanning mechanisms may be considered as either proximal or distal to the light source [20]. Proximal scanners are placed in the illumination pathway upstream of the fiber and are used with a fiber bundle. This configuration offers the benefit of separating bulky scanners from a miniaturized imaging head and typically includes cascaded galvanometer-mounted scanning mirrors, scanning the beam across the proximal end of a fiber bundle. Distal scanners are placed on the fiber side distal to the light source and usually scan illumination from a single fiber over the specimen. Here, the 2D scanning can be performed by a MEMS mirror which pivots in two angular directions or by maintaining the resonant vibrating of the fiber extremity via an attached actuator or cantilever.

After comparing the different configurations of OCT probes, Figure 7 summarizes the characteristics made for each criterion of the probe classification proposed earlier. Here, the colorful path corresponds to the optimal design of an OCT probe applied for gastrointestinal tract evaluation [38].

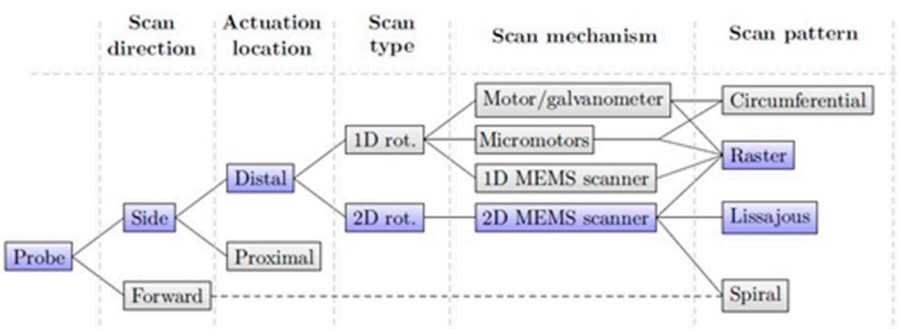

**Figure 7.** Order of criteria of classifications of the endoscopic probe configurations.

## 4. Electrostatic MEMS Scanning Mirrors for OCT

### 4.1. Principles of Electrostatic Microactuators

The principle of electrostatic actuation is based on Coulomb's law, using the attraction of two oppositely charged plates. Electrostatic actuation is currently the predominant method used for MEMS scanners because the capacitive actuators draw very little current, therefore requiring low operating power despite the need for relatively high applied voltages. An electrostatic torsional micromirror is in rotation when a driving voltage is applied between the fixed and movable electrodes. The mirror rotates an angle θ about the torsion axis until the restoring and electrostatic torques are equal. The torque is given by:

$$T_c(\theta) = \frac{V^2}{2} \frac{\partial C}{\partial \theta} \tag{3}$$

$$T_r(\theta) = k\theta, \tag{4}$$

where $V$ is the driving voltage, $C$ the capacitance of the actuator, and $k$ is the spring constant.

For a simple parallel plate actuator, the capacitance is given by:

$$C = \frac{\varepsilon_0 A}{g},$$

(5)

where $\varepsilon_0$ is the permittivity of free-space, $A$ is the surface of electrode, and $g$ is the gap between the electrodes.

Figure 8 shows the schematic of two main types of electrostatic actuators: the parallel plate actuator [39] and the comb-drive actuator [40,41].

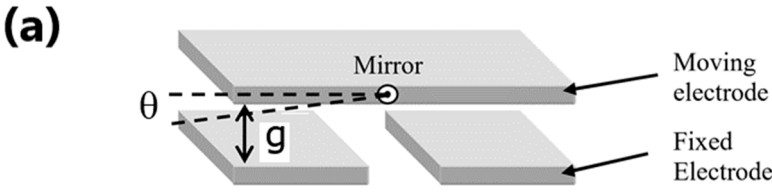

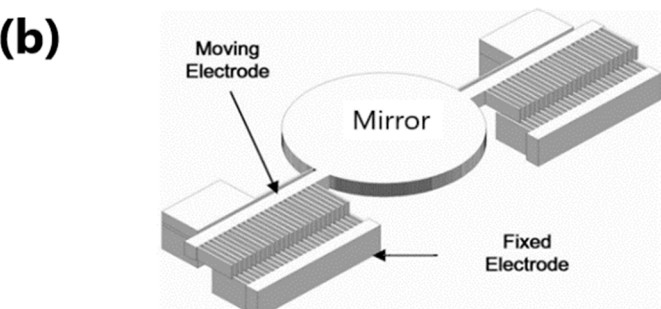

**Figure 8.** Two main architectures of electrostatic actuation: (**a**) parallel-plate actuator and (**b**) comb-drive actuator.

With a parallel plate gap closing actuator, the zone of the electrode overlap is mainly the area of the fixed electrode. Thus, the gap is a function of the rotation angle. Here, there is a tradeoff as the initial gap distance needs to be large enough to generate the scan angle, but small enough for a reasonable driving voltage. The linear scan range is limited by the pull-in effect to around 40% of the maximal mechanical scan angle [42]. Electrostatic actuators are relatively easy to fabricate by micromachining technologies. Parallel-plate actuators employ surface micromachining, often based on polysilicon with sacrificial oxide, on electroplated metal with sacrificial organic layer or sputtered metal with a sacrificial organic layer. Comb-drive actuators are typically fabricated on Silicon-on-Insulator (SOI) substrates, ensuring a relatively simple fabrication process and easy thickness control of micromechanical structures.

*4.2. Examples of Electrostatic OCT Probes*

A series of endoscopic OCT probes based on electrostatic actuation have been proposed, using 2D MEMS scanners that scan in two axes [43–46] and employing a 2D gimbal-less vertical comb-drive structure. An interesting example of an OCT endoscopic MEMS scanner for high resolution OCT with angled vertical comb-drive actuators has been proposed by Aguirre et al. [47] at MIT. Figure 9a shows the SEM photography of this MEMS scanner. The microscanner uses a torsional beam and includes a gimbal-mounting MEMS mirror to scan a dual axis, combining the scan's x and y axes with a single pivot point. The actuated mirror provides ±6° angular scanning at over 100 V of driving voltage. Here, a silicon micromirror is suspended inside a gimbal frame by a pair of polysilicon torsion springs. The scanning mirror has a circular aperture with a diameter of 1 mm within the footprint size of $3 \times 3$ mm$^2$.

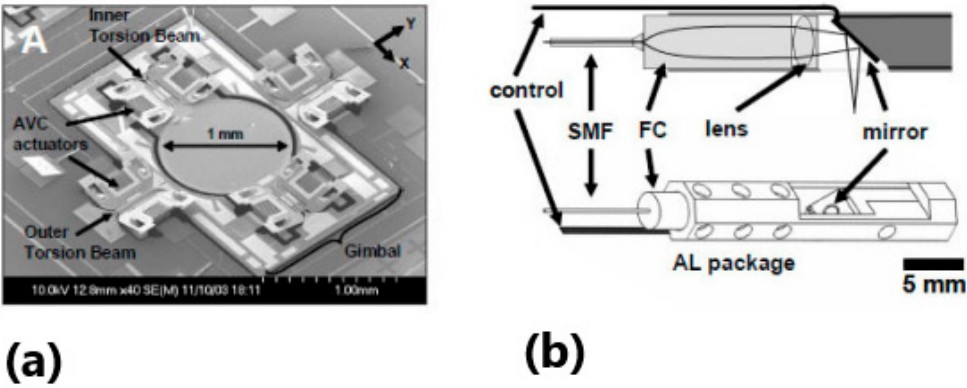

**Figure 9.** SEM photograph of 2D MEMS scanner in (**a**) and the schema of MEMS catheter packaging in (**b**) (Figures from [47]).

Figure 9b shows the schematic of OCT catheter packaging. The 2D MEMS scanner is inclined at 45° and directs the beam in a side scanning configuration, orthogonally to the endoscope axis. The post-objective scanning eliminates off-axis optical aberration encountered with pre-objective scanning schemas. The endoscope head is 5 mm in diameter and 2.5 cm long. The optics include a graded index fiber collimator followed by an AR-coated achromatic focusing lens and which produces a beam spot diameter of 12 μm. Figure 10a represents the resonance characteristics of the MEMS scanner. The mirror resonance is 463 Hz and the gimbal axis resonance is around 140 Hz. Resonant operation of the mirror offers high speed raster scanning for en-face microscopy. The MEMS OCT probes were demonstrated in 3D high resolution OCT imaging. The OCT catheter was combined with an OCT device employing a commercial femtosecond Nd:glass laser, centered on the wavelengths of 1.06 μm with a bandwidth of more than 200 nm. The light source was coupled with a fiber-optic interferometer. The sample was measured with ~2000 axial scans per second. The interference signal was first amplified, filtered, and then demodulated by the detection block including a 12-bit, acquisition card, and a 5 MHz A/D converter and then processed by PC computer. The obtained axial resolution of images was <4 μm in tissue, while transverse resolution was equivalent to the focusing spot of 12 μm. Imaging was performed at a rate of 4 frames/s over a 3D field of view of $1.8 \times 1 \times 1.3$ mm$^3$ with $500 \times 500 \times 1000$ pixels. Figure 10b illustrates an example of a 3D image that represents a volume data set from the hamster cheek pouch acquired in vitro.

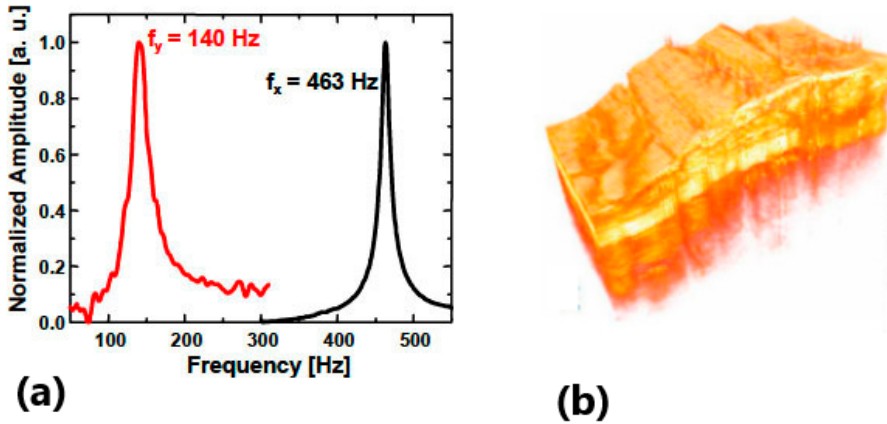

**Figure 10.** Resonance characteristics of the MEMS scanner (**a**); and 3D OCT image acquired in vivo set from the hamster cheek pouch (**b**) (Figures from [47]).

According to the schema of catheter packaging in Figure 9b, the OCT probe is limited in size by the footprint of the MEMS mirror—this is the main limit in the miniaturization of such OCT devices. High driving voltage is also an issue for in vivo endoscopy. The acquisition of data is made in open-loop mode, which introduces a possible lack of precise control of the MEMS mirror. This may degrade the scanning trajectory stability and data reproducibility. Addressing this concern will require one to produce more sophisticated MEMS technology, including the possibility of close-loop control detection of mirror motion. An alternative based on the use of in-situ capacitive detection will be described in the next paragraph.

The interest to miniaturize OCT probes is not only limited to the endoscopic configurations. Existing bulk microscopes or fiber optics devices for early diagnosis of cancer are expensive and are only affordable at the hospital; thus, they are not sufficiently used by physicians or cancer specialists as an early diagnosis tool. Significant reduction of system cost and size can be achieved by use of opto-mechanical components, fabricated by micromachining. In 2016, an OCT microsystem including an active $4 \times 4$ array of spectrally tuned Mirau interferometers, including an electrostatic vertical comb-drive actuator carrying the array of reference mirrors, was proposed for dermatology applications [48]. The architecture of an active Mirau interferometer is shown schematically in Figure 11a. To perform OCT measurements, the device is incorporated within an experimental setup including a swept-source laser (center wavelength: 850 nm, swept range: 50 nm) and a high speed smart camera.

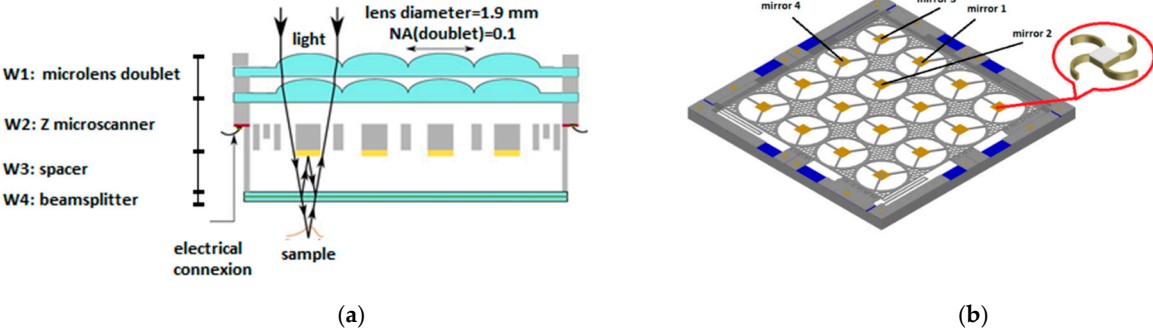

(**a**)　　　　　　　　　　　　　　　　　　　　(**b**)

**Figure 11.** Cross-sectional view of the multichannel "active" Mirau microinterferometer (**a**) with a focus on a 3D view of an electrostatic vertical microscanner with a $4 \times 4$ array of suspended reference micromirrors (**b**).

A Mirau interferometer includes a series of vertically stacked components: a doublet of microlens matrices, a vertical comb-drive actuator, a spacer, and a planar beam splitter plate. The diameter of an individual microlens is 1.9 mm and the array pitch is 2 mm, while the equivalent focal length of a microlens doublet is 7.44 mm and the numerical aperture (NA) is 0.1. The assembly of two glass microlens arrays is made by anodic bonding. The axial resolution of OCT imaging is 6 μm, while the transverse resolution is limited to 6 μm. The depth of penetration is 0.6 mm. The key element of a Mirau interferometer is the vertical microscanner W2. The microscanner is designed for generating a vertical displacement of a large platform with a $4 \times 4$ array of reference micromirrors of the Mirau interferometer, as shown in Figure 11b.

The vertical motion of the whole $4 \times 4$ array of reference micromirrors at the resonance frequency can be controlled precisely by an in-situ differential position sensor measuring the variation of capacitance due to the comb-drive displacement [49]. The vertical actuation of reference mirrors leads to a phase-shifted imaging that enables rapid measurement of the amplitudes and phases of interference signal and improves the signal to noise ratio and sensitivity. The size of an individual micromirror, suspended by a system of spider legs, is $400 \times 400$ μm$^2$. The array of micromirrors is vertically aligned with the lenses, forming an $8 \times 8$ mm$^2$ structure. The resulting imager covers the same area of $8 \times 8$ mm$^2$ of the sample, reconstructing the topography in a continuous way by stitching together

$4 \times 4$ single-channel interferograms via a system of actuators shifting mechanically the entire Mirau-array over the overlapping region. The 3 mm thick spacer W3 in silicon adjusts the position of lens focus from the planar beam splitter plate. The beam-splitter W4 has a transmission-reflection ratio of 70/30. Figure 12a shows the assembled Mirau interferometer mounted on the PCB (printed circuit board). The footprint of this chip is $15 \times 15$ mm$^2$, whereas the overall thickness is about 5 mm.

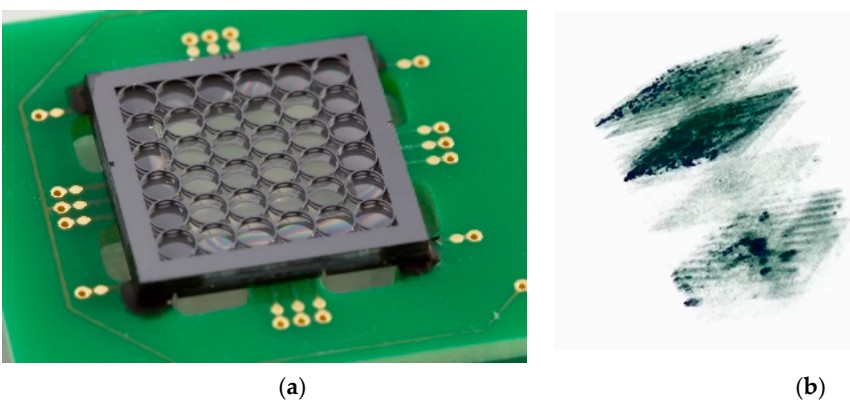

(**a**)　　　　　　　　　　　　　　　　　　　　　　　　(**b**)

**Figure 12.** PCB-mounted chip of a Mirau interferometer in (**a**); and 3D swept-source OCT image of onion slices in (**b**).

The original A-scan includes several parasitic terms such as the autocorrelation terms due to the beam splitter, reference mirrors images, mirror replica images, and DC noise term, making the interpretation of true OCT signal difficult. Help to the use of four-frame phase shift algorithm all these signals are removed, improving both the signal-to-noise ratio and the measurements range. Figure 12b shows a volumetric $300 \times 300 \times 600$ µm$^3$ OCT image of an onion slice, where the microscopic structure is visible. The sensitivity of this image is in the range of 80 dB.

## 5. Electromagnetic MEMS Scanning Mirrors for OCT

### 5.1. Principles of Electromagnetic Microactuators

Electromagnetic MEMS actuators are driven by Lorentz force [50]. In this case, a current-carrying conductor is placed in a static magnetic field. This field produced around the conductor interacts with the static field to produce a force. In an electromagnetic actuator, including the flexure beams and a moving mirror plate, the module of Lorentz force is expressed as:

$$F = BIL \sin\theta, \tag{6}$$

where $B$ is magnetic flux density of the magnetic field, $I$ is the current flowing through the beam, $L$ is the beam length, and $\theta$ is the angle between the current and the magnetic field.

There are numerous variations on the architecture of the electromagnetic actuators: permanent magnets interacting with an external field, permanent magnets interacting with current-carrying coils, and current carrying conductors interacting with an external field. A common advantage is the relatively high generated force. The main drawbacks are the high-power dissipation as well as a complex fabrication, including severe materials challenges, and difficulty to miniaturize the micromirrors because of the use of external bulk magnets. Figure 13 shows the schematic design of a 2D electromagnetic scanner where the Lorentz force interaction is generated between the micro-coil integrated on the scanner gimbal and the permanent magnets located outside of the scanner.

To decrease the size of electromagnetic actuators, a combined electrostatic/electromagnetic 2D scanner has been developed by Microvision for retinal scanning displays [29]. Such a mixed configuration employed electromagnetic actuation to move the outer frame, which provides the slow vertical axis, while electrostatic actuation was used for the inner mirror axis, which provides the fast horizontal axis.

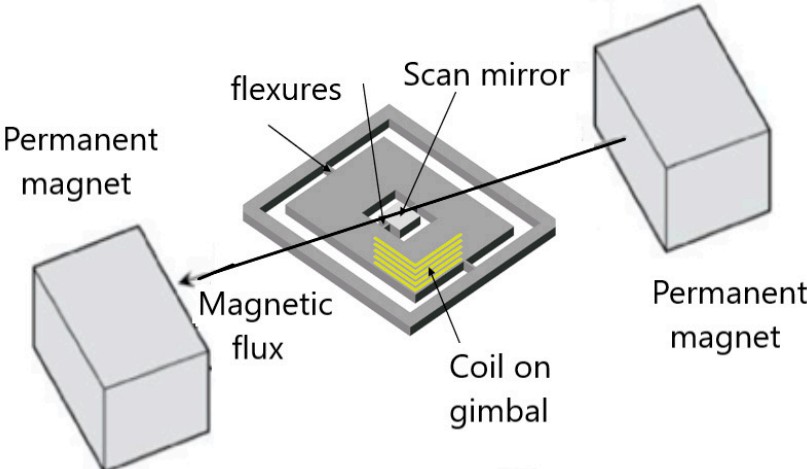

**Figure 13.** 2D electromagnetic actuator generating the Lorentz force between the micro-coil integrated on the scanner gimbal and the two permanent magnets.

*5.2. Examples of Electromagnetic OCT Probes*

Magnetically actuated scanning mirrors made by micromachining were demonstrated by Judy et al. [51]. The Olympus Company developed one of the first 1D electromagnetic MEMS scanners for confocal microscopy [52], while an early 2D electromagnetic MEMS scanner was proposed by Asada et al. [53]. This category of MEMS scanning mirrors replaced the electrostatic actuators in situations where it was necessary to increase the scanning range or lower the driving voltage. Serious efforts were performed to make the commercialization of MEMS electromagnetic scanner OCT probes easier for clinical applications. To overcome the fabrication complexity of earlier electromagnetic actuators, for example, a flexible 2-axis polydimethylsiloxane (PDMS)-based electromagnetic MEMS scanning mirror was developed [54]. The size of such a MEMS scanner remained relatively big ($15 \times 15 \times 15\,\text{mm}^3$).

An interesting electromagnetic scanning actuator for OCT imaging was demonstrated by Kim et al. [55]. Figure 14 shows the photographs of a two-axis gimbaled mirror with folded flexure hinges. The rotation of the mirror is possible in two axes along the flexures: one inner axis and one orthogonally placed outer axis. Resonant frequencies for the inner and outer axes were 450 Hz and 350 Hz, respectively. To generate the magnetic actuation, a permanent magnet glued to the backside of the mirror plate and a pair of coils is placed inside the probe body for each scan direction. The mirror plate is $0.6 \times 0.8\,\text{mm}^2$, with a device footprint of $2.4 \times 2.9\,\text{mm}^2$.

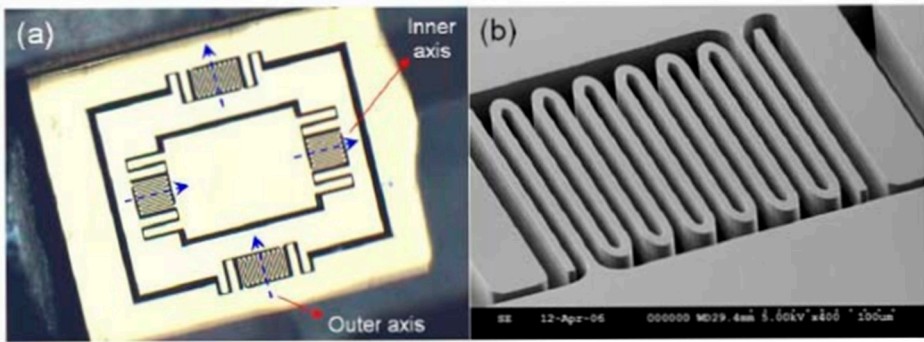

**Figure 14.** Photograph of electromagnetic micro-scanner in (**a**) and a SEM image of folded flexure hinges in (**b**) (Figures from [55]).

Figure 15a shows a schematic of the assembled catheter packaging. The light source is pigtailed by a single mode optical fiber, delivering via a GRIN lens a focused beam

redirected by the MEMS mirror on the sample, which is then scanned. Scattered light from the sample returns through the same optical path and is collected by the pigtailed GRIN lens. A glass window with an AR (anti reflective) coating protects the MEMS scanner and eliminates the back reflections. The catheter package has a 2.8 mm diameter and a 12 mm length. The MEMS scanner is fabricated from a SOI wafer with a 50 μm thick device layer on a 350 μm thick handle layer, including a 1 μm thick oxide box layer. Magnet layers are composed of small NdFeB magnets, each one measuring $0.6 \times 0.8 \times 0.18$ mm$^3$. Figure 15b represents optical angles of the MEMS scanner in both inner and outer axes as a function of the driving voltage. An optical scan angle of about $\pm 30°$ was obtained with $\pm 1.2$ V and $\pm 4$ V driving voltages for the inner and outer axis, corresponding to 50 mA and 100 mA current, respectively.

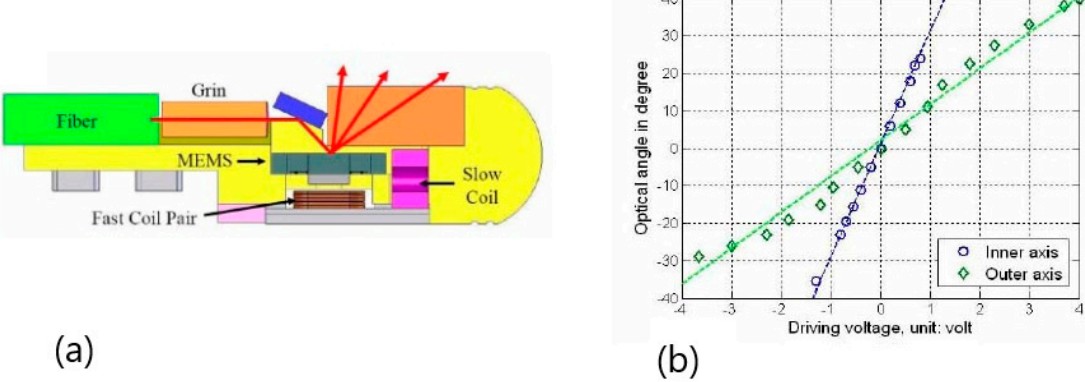

(a)  (b)

**Figure 15.** Schematic of the assembled catheter packaging in (**a**) and optical angles of the MEMS scanner for both axes vs. the driving voltage in (**b**) (Figures from [55]).

The light refracted by the protection window produces slight nonlinearity in the deflection angle due to thickness variations of the window. Spurious vibrations are observed for large scan angles at the mirror resonant frequency. To avoid these effects, the working scan angle was reduced to $\pm 20°$ optical angle for the inner axis and less than $\pm 30°$ optical angle for the outer axis. Optical resolution was estimated to be 5 μm. In vivo 3D endoscopic imaging of tissues was made by combining the two-axis scanning catheters and the multifunctional SD-OCT system. 3D images of a fingertip were acquired at 18.5 frames/s, with the scan performed with voltages of $\pm 2.8$ V and $\pm 0.8$ V applied on the inner and outer axis, covering $1.5 \times 1$ mm$^2$ lateral scan range and consuming 150 mW of power.

Figure 16 shows a 3D OCT image of fingertip tissue where the fingerprint orientations are visible.

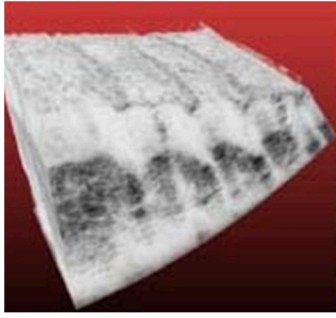

**Figure 16.** 3D OCT image of fingertip tissue where the fingerprint orientations are visible (Figure from [55]).

Watanabe et al. [56] have demonstrated an electromagnetic MEMS scanner for OCT imaging. The schematic diagram of a fabricated mirror scanner is shown in Figure 17. The device includes a 0.2 mm thick silicon frame carrying the micromirror with its actuator,

a printed circuit board, and a magnet holder with a magnet inside. The minimum size of the magnet is $6 \times 6 \times 5$ mm$^2$. A metal coated $1.8 \times 1.8$ mm$^2$ mirror and two y-scan coils are formed on the y-frame in the center of the silicon frame. The folded y-scan beams are supported by an x-frame. Two x-scan coils are formed on the x-frame. All the coils have a dimension of $2 \times 2$ mm$^2$. The folded x-scan beams are supported by an external fixed frame. When a current is passed through the y-scan coils, the mirror deflects in the y direction. When a current is passed through the four x-scan coils, the mirror tilts in the x direction. Thus, the light beam can be 2D steered. The entire microscanner is mounted on a $15 \times 6$ mm$^2$ PCB, which is fixed on the magnet holder.

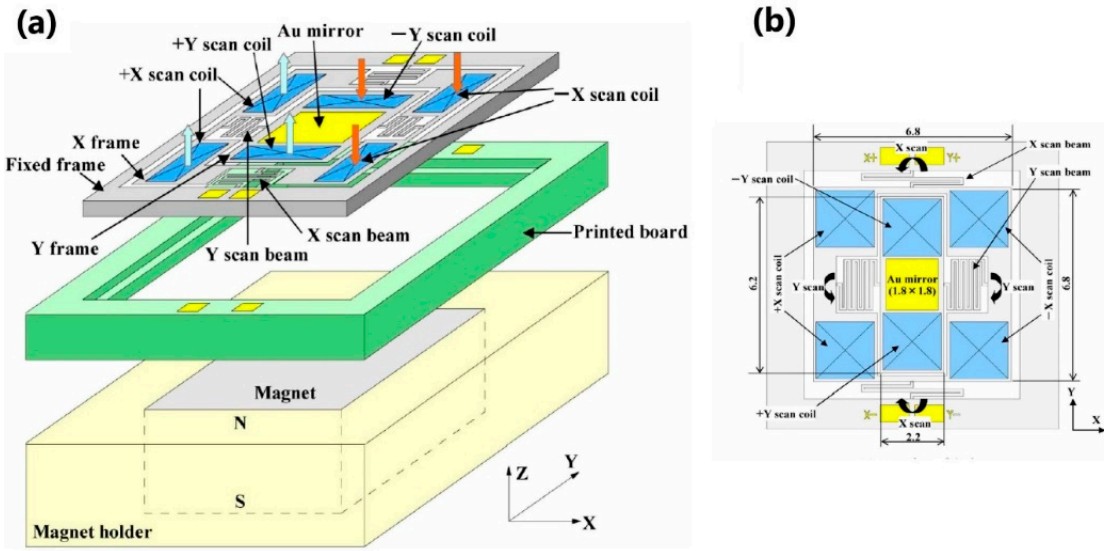

**Figure 17.** Architecture of the MEMS electromagnetic scanner: (**a**) device schema and (**b**) the top view of the mirror actuator (Figures from [56]).

The MEMS microscanner was placed in a Fourier domain OCT setup including a SLD light source operating at 1.55 μm and a fiber optic Michelson interferometer. OCT images of human fingers were obtained, as shown in Figure 18.

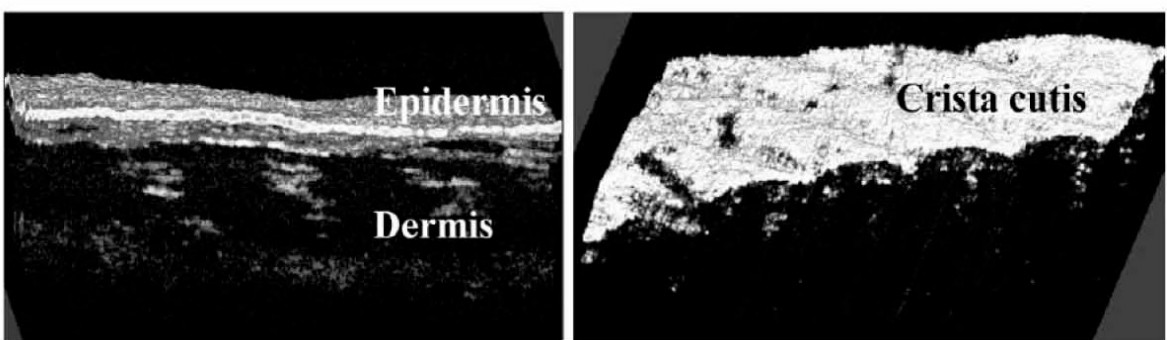

**Figure 18.** $3 \times 3$ mm$^2$ OCT images of human fingers: the epidermis including stratum corneum and the crista cutis (Figures from [56]).

The scanners discussed here demonstrate that one of the main drawbacks of electromagnetic mirrors is that an external magnet is required for actuation. Such magnet technology is often not compatible with the process flow of the micro-scanner. A bulky magnet reduces the potential of miniaturization of the probe. Another drawback concerns the relatively high-power consumption of electromagnetic scanners. Finally, the

relative complexity of the fabrication process and high costs are a bottleneck for fast clinical translation of the electromagnetic MEMS scanners.

## 6. Electrothermal MEMS Scanning Mirrors for OCT

### 6.1. Principles of Electrothermal Microactuators

The principle of an electrothermal actuator is based on the Joule heating and thermal expansion principles. The actuation uses the balance between the thermal energy generated by an electrical current and the heat dissipation through the actuator structure [57,58]. The three categories of electrothermal actuators are bimorph actuators, chevron actuators, and hot-and-cold-arm actuators. The structure of the more popular bimorph actuator contains two layers of materials with different coefficients of thermal expansion $\alpha_T$ (CTE). A metallic heater is sandwiched between the active materials, as shown in Figure 19. The injection of electrical current within the heater layer generates the Joule effect in the active materials and produces the deflection angle. Equation (7) shows that the mechanical strain of material $\varepsilon$ is directly proportional to the temperature change $\Delta T$:

$$\varepsilon = \alpha_T \Delta T. \tag{7}$$

The curvature of generated mechanical bending can be approximated as

$$R = \frac{t_1 + t_2}{\varepsilon_1 - \varepsilon_2}, \tag{8}$$

where $t_{1,2}$ is the thickness of active layers and $\varepsilon_{1,2}$ represents the thermal strain of active layers. A wide range of active materials can be used. Thus, the CTE of silicon is $2.6 \times 10^{-6}$/K, while that of $SiO_2$ is $0.35 \times 10^{-6}$/K, aluminum CTE is $25 \times 10^{-6}$/K, and the CTE of polyimide from Amoco Ultradel 1414 is $191 \times 10^{-6}$/K.

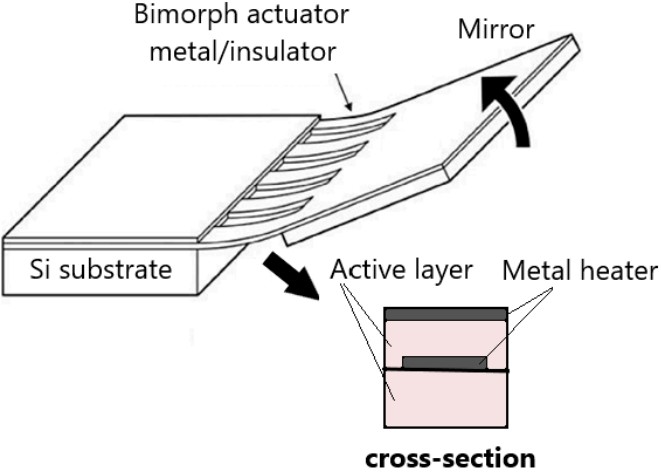

**Figure 19.** Principle of electrothermal actuation.

### 6.2. Examples of Electrothermal OCT Probes

In electrothermal actuators, the actuation force is typically larger than that of electrostatic and electromagnetic actuators. They benefit also from high fill factor. Such characteristics make the electrothermal bimorph-based MEMS scanners very suitable for miniaturizing of OCT probes for endoscopic applications. An example of such a probe, based on a 2D thermal bimorph micromirror, has been developed by Sun et al. [59]. The resulting microscanner includes a mirror suspended without the gimbal by four actuators on its four sides. The actuator is based on three Al/SiO$_2$ bimorph beams connected in series with a Pt heater embedded for electrothermal actuation and two rigid suspended silicon frames. This design is called lateral-shift-free (LSF) LVD design [60].

More recently, the architecture of a swept-source OCT endomicroscope has been demonstrated, including a mirror scanner with a similar mechanism of actuation [61]. The OCT probe contains a spectrally tuned single-channel Mirau micro-interferometer, integrated with a two axis MEMS electro-thermal micro-scanner. This optical microsystem operates in the side-imaging mode.

Figure 20 shows the schematic of an OCT microsystem where a $4.7 \times 4.7 \times 5.3$ mm$^3$ Mirau microinterferometer is shown inside the blue square. The size of the microscanner external frame is $4 \times 4$ mm$^2$, with a mirror diameter of 1 mm. The monolithically integrated Mirau microinterferometer [62] includes a silicon base for a GRIN lens assembly port, a glass wafer with a reflowed focusing lens with a focal length of 9 mm, a reference micromirror, a silicon separator, and a beam splitter plate. The GRIN lens generates a collimated light beam with a diameter of 1 mm, which illuminates the plano-convex Mirau glass lens of 1.9 mm diameter [63]. The glass beam-splitting plate divides the converging beam into a reference beam and a scanning beam. A silicon separator ensures the position of the beam splitter plate at half of the focal length of the focusing lens. The reference beam is back-reflected from the 150 μm reference micromirror at the backside of the focusing lens, whereas the scanning beam is directed by the MEMS scanner towards the sample to be measured. The schematic of a MEMS scanner, based on a two-axis MEMS electrothermal micro-mirror, is shown in Figure 21a, while the image of a MEMS scanner assembled on top of a Mirau interferometer is shown in Figure 21b [58].

The inner mirror plate is connected to a rigid frame via a pair of torsional bars in two diametrically opposite ends located on the rotation axis. A pair of electrothermal bimorphs generates a force onto the perpendicular free ends of the mirror plate in the same angular direction. An array of electrothermal bimorph cantilevers deflects the rigid frame and a mechanical stopper maintains the position of the mirror inclined at 45° from the optical axis. The performed scans reach large mechanical angles of 32° for the frame mirror and 22° for the in-frame mirror. Figure 21c,d shows the deflection amplitude of the micromirror versus the frequency response of the pitch-axis and roll-axis, respectively. Here, a small coupling between both the axes is observed. The fabricated micromirror has a mechanical resonant frequency around 1200 Hz for both axes.

Figure 22a shows the real-time reconstruction of a "femto-st" pattern, obtained by Lissajous imaging at a sampling frequency of 1 MHz [64]. The acquisition of data was performed in open-loop mode. The pattern lines of 30 μm-wide are well-resolved.

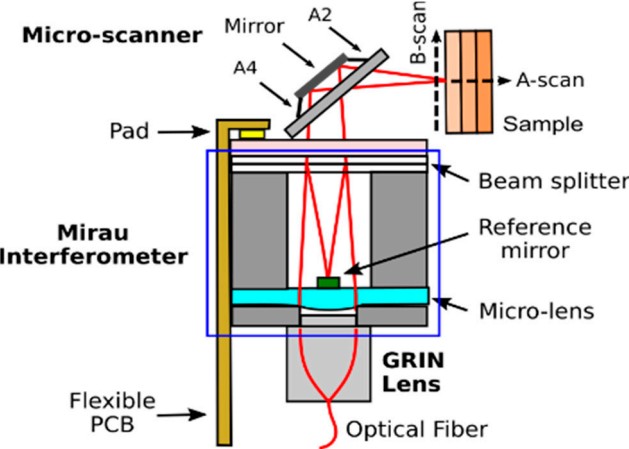

**Figure 20.** Schematic diagram of the MOEMS (Micro-Opto-Electro-Mechanical Systems) probe.

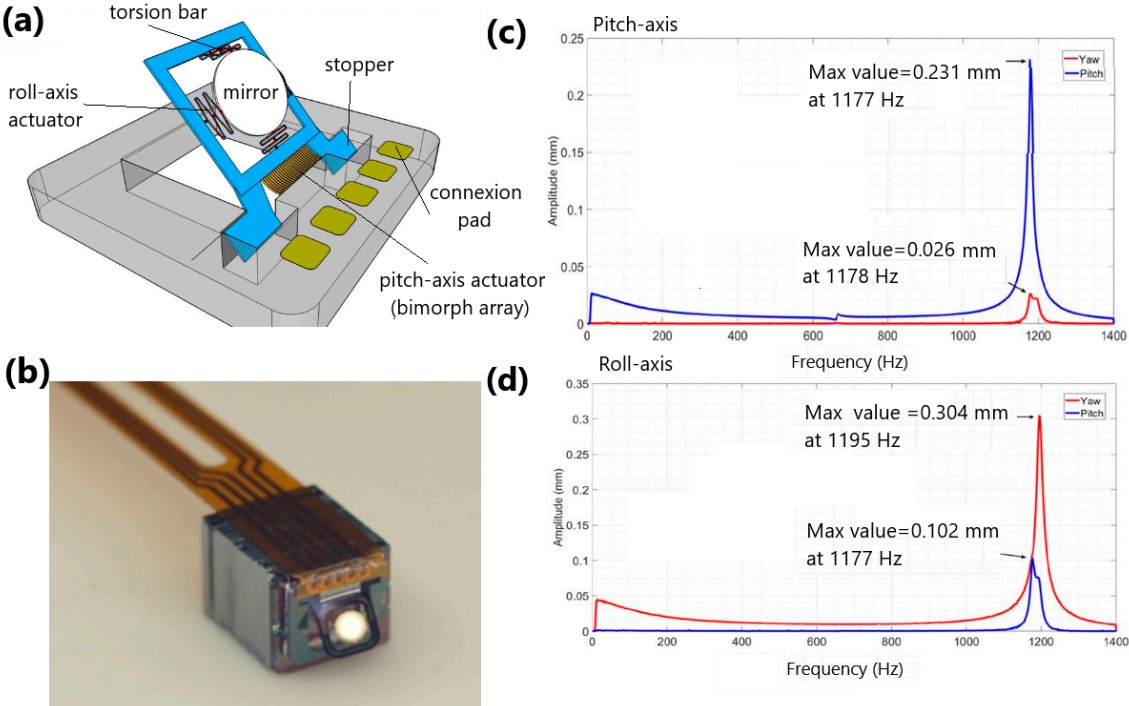

**Figure 21.** Electrothermal mirror: (**a**) schematic and (**b**) microphotograph of a MEMS scanner assembled on top of a Mirau interferometer. The frequency response for inner pitch axis (**c**) and roll-axis (**d**).

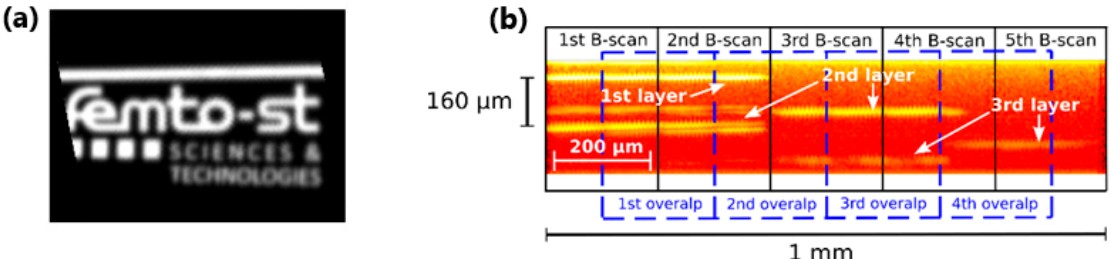

**Figure 22.** Reconstruction of a "femto-st" pattern in (**a**) and B-scan of a multilayer glass sample in (**b**).

The complete OCT probe is connected to the illumination and detection blocks by a single-mode optical fiber. The system is illuminated by a swept source with a central wavelength of 840 nm and A-scan frequency of 110 kHz [65]. The OCT images were obtained from a sample made of three cover glasses, each 160 μm thick. Figure 22b shows the averaged B-scan images of this sample.

Circumferential scanning for endoscopic OCT with MEMS electrothermal mirrors was demonstrated in 2018 by S. Luo et al. [66]. This microscanner uses a circular array of six scan units, including electrothermal MEMS mirrors and C-lens collimators with a focal length greater than 10 mm, as shown in Figure 23. This compact microscanner presented a chip size of $1.5 \times 1.3$ mm$^2$. Each C-lens and a single-mode fiber are packaged inside a glass tube with a diameter of 1.4 mm. The full circumferential scans have been demonstrated with individual micromirrors scanning up to 45° at a voltage of less than 12 V.

Figure 24a shows the MEMS mirrors with a $0.5 \times 0.5$ mm$^2$ mirror plate. Four bimorph actuators are placed symmetrically at the four sides of a central mirror plate. Each bimorph actuator consists of three pairs of double S-shaped bimorph (Al/SiO2) beams, sandwiched with a heater layer made of a thin film of Ti/TiN. A mechanical scan angle of 13° is achieved, resulting in a $\pm 26°$ optical scan angle or a 52° field of view (FOV).

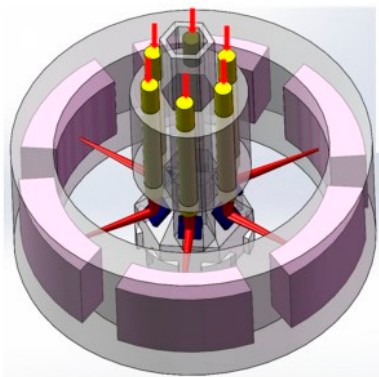

**Figure 23.** The schematic of MEMS OCT probe (Figures from [66]).

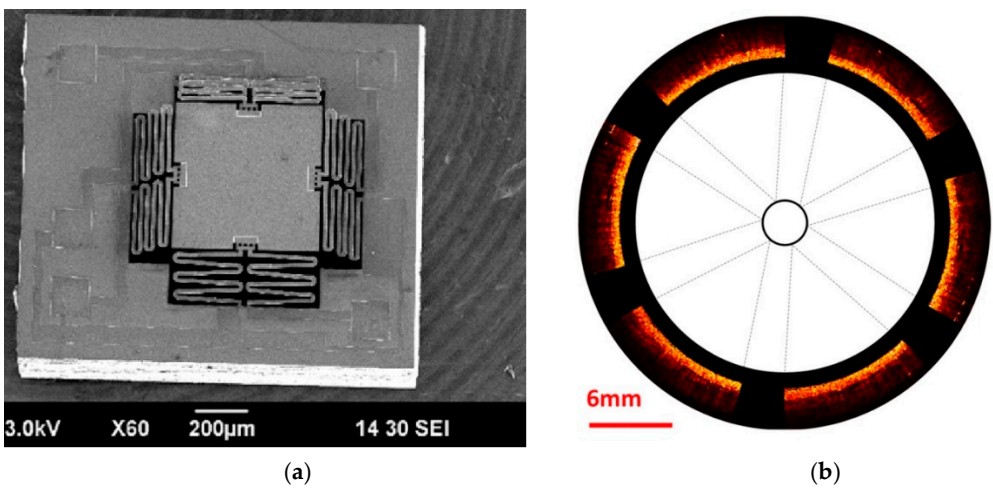

| (**a**) | (**b**) |

**Figure 24.** An SEM image of the MEMS mirror in (**a**) and the imaging sample of a swine's small intestine covering a glass tube in (**b**) (Figures from [66]).

Configured with a swept-source OCT setup, this MEMS array-based circumferential scanning probe was applied to image a swine's small intestine wrapped on a 20 mm-diameter glass tube, as shown in Figure 24b. The OCT imaging result shows that this new MEMS endoscopic OCT has promising applications in large tubular organs.

An alternative solution is to move the objective lens directly by an actuated microstage. L. Wu et al. [67] demonstrated a tunable microlens scanner, operating at a 880 nm wavelength. This microscanner uses a lateral-shift-free electrothermal bimorph actuator, carrying a 1 mm diameter glass rod lens moving at a resonance frequency of 79 Hz. Later, L. Liu et al. [68] developed another electrothermal MEMS microlens scanner, moving a 2.4 mm plano-convex glass lens with a maximum travel range of 400 μm and a resonance frequency of 24 Hz.

A more compact microlens scanner was developed in 2020 by L. Zhou et al. [69] where the actuation mechanism is based on a single serpentine inverted-series-connected (ISC) electrothermal bimorph actuator carrying a microlens. The shape of the entire microlens scanner is circular, with an outer diameter of 4.4 mm and a clear optical aperture of 1.8 mm, as shown in Figure 25a. The microstage includes a ring-shaped frame and eight sets of bimorph actuators. It is loaded with a 2.4 mm plano-convex glass microlens on its ring-shaped frame. The microlens weight is about 8 mg. Figure 25b shows the S-shaped form of an ISC electrothermal actuator where the round hinges are used to connect all ISC structures, minimizing the residual stresses during motion. The resonant frequency of the MEMS microstage loaded with a lens reaches 140 Hz, which is acceptable for endomicroscopic imaging tasks.

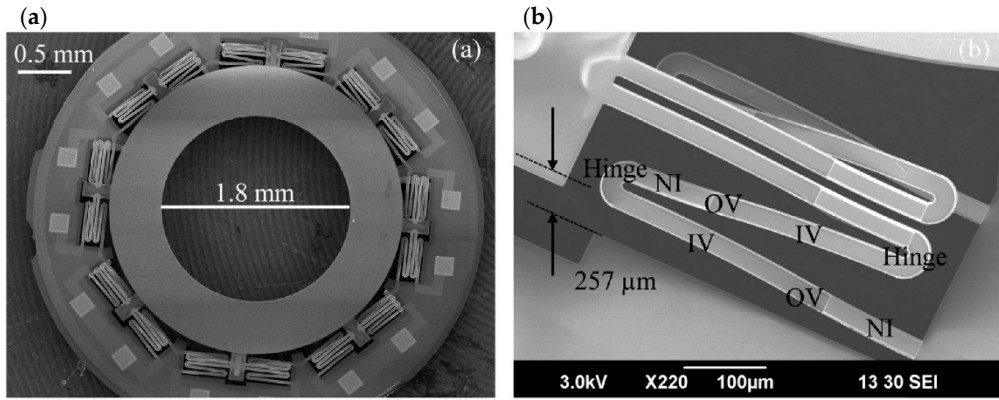

**Figure 25.** SEMs of the microlens scanner: (**a**) top view of complete structure and (**b**) focus on one of eight inverted-series-connected (ISC) actuators where the silicon beam is suspended 257 μm above the outer silicon frame (Figures from [69]).

## 7. Discussion on the Microscanner Design and Conclusions

The replacement of galvanometer mirrors for OCT beam scanning by 2D MEMS scanning mirrors converted the bulk microscope to a compact and light device. However, galvanometers demonstrate better performance because they operate in a closed loop, using position feedback to correct the drive waveform instead of an open loop for MEMS scanning *mirrors. When compared to MEMS mirrors, galvanometers are higher cost and relatively larger.

This paper demonstrated that current MEMS scanning technologies have advantages and limitations compared to galvanometers. The use of MEMS scanning mirrors in OCT and other biomedical applications reduces the complexity of scan control and offers a lower cost scanner. To build minimally invasive endoscopic probes, scanning micromirrors are required to be compact (<5 mm). The size of the scanning mirror is a crucial parameter because the micromirror dimension should be larger than the laser spot size as well. High speed and large transverse scans can also be achieved, which enables real-time in vivo imaging and a large field of view, respectively. To design the ideal MEMS scanner for OCT, we need to consider the scan angle, the resonance frequency, dynamic, as well as static mirror flatness, and good resonant mode separation. The choice of these parameters influences the image quality, the desired resolution, and the presence of image distortions due to aberrations from the scanning mirror and objective lens and defines the limits for scan speed and total image size.

As we demonstrated in Section 2, the specifications of a MEMS mirror determine the number of resolvable spots and the OCT B-scan rate. Increasing the size of the MEMS mirror or scan angle would increase the number of resolvable spots. The repetition frequency of a B-scan is limited by the resonance frequency of the MEMS mirror. Such resonance frequency is defined by the mirror inertia and is proportional to the inverse square of the mirror diameter. Generally, we would select a MEMS scanner that achieves the largest angle and the highest operating speed if large areas are to be imaged quickly. In clinical applications, the minimal line scan lengths must cover approximately a range from 1 mm to 2 mm. To maximize the imaging speed, the solution is to perform the scan at the resonance frequency of the MEMS mirror. However, the scan operation performed at the resonance frequency is a source of image distortions produced by parasitic vibrations of scanner axes. To overcome the parasitic vibrations, the B-scan frequency must be fixed below the resonance frequency of the MEMS mirror.

The size of scanning micromirrors included in the OCT probe strongly influence the scanning performances and specifications of the OCT probe. Smaller diameter scanning mirrors facilitate both the integration of the probe within the standard endoscope and probe guidance in the internal organs. Increasing the mirror diameter reduces the resonance frequency, resulting in a slower scan repetition frequency, limiting the number of B-scans.

In this case, if the suspension is stiffened to maintain a given speed, the torque available from the actuator must be increased. The thickness of the MEMS device is also crucial in the definition of micromechanical features of the MEMS device. Thus, the thicker electrodes of a comb-drive actuator improve the electrostatic force without affecting the deformation of the scanning micromirror. A thinner air gap of the comb-drive, obtained via surface micromachining, would decrease the high driving voltage (often ~100 V), making the device safer for patients. In microactuators where the moving part is a thin membrane, thinner membranes improve the motion range because of the increase of actuator deformation. In conclusion, the advantages of smaller MEMS scanning mirrors include the smaller mass, lower stiffness, and higher imaging speed. All of these parameters must be carefully considered when choosing the appropriate MEMS scanning mirror for a specific application of OCT imaging. Many parameters of the MEMS device and optical probe also must be selected during design, fabrication, or assembly. Others can be adjusted during the OCT experiment.

Table 1 compares qualitatively the characteristics of MEMS scanning micromirrors for three types of actuation mechanisms, based on the literature review [70]. This study focuses on the highest performing MEMS scanning mirrors designed for miniaturized displays and optical imaging. The typical diameter of scanning micromirrors ranges from 0.5 mm to 3.5 mm. However, an average diameter of 1 mm is observed for all categories of MEMS scanning mirrors. Average values of micromirror mechanical specifications for all types of actuators are compared. Analyzing the range of motion for electrostatic microactuators, we can see that the average performance of the vertical comb actuators is 14° for an average piston motion of 94 μm, which is better than that of linear comb-drives (8.5° for piston motion of 29 μm). The average performance of electromagnetic actuators is 15° and 5 μm for angular motion and piston motion, respectively. Finally, the electrothermal actuators present the best performances in both rotational motion and out-of-plane motion. Here, the average angular motion is about 27° and 280 μm for average piston motion. An important characteristic to be analyzed is the resonance frequency for each category of microactuators. The resonance frequency of electrothermal and electromagnetic micromirrors is in the range from 100 Hz to 1000 Hz, while the resonance frequency of the comb-drives stays within the range of 150 Hz to 10,000 Hz. The required response time for dynamic systems is about 5 ms for scanning micromirrors implemented in OCT probes.

Table 1 shows that each actuation principle has advantages in some aspects while having disadvantages in others. Electrostatic actuation has a fast response and the lowest power consumption, but it requires large driving voltage, which may not be safe for endoscopic applications. In addition, the electrostatic scanners have strong nonlinearities, limiting the MEMS displacement. Here, the electrostatic torque is a function of $V^2$, not linearly varying with the actuation voltage. This can result in a distortion of the scan pattern when driving with linearly ramped voltages. Several approaches of linearization have been proposed to eliminate the distortion, improving the linearity of scan patterns [71]. Finally, an additional advantage for electrostatic comb-drive actuators is the easy and standardized micromachining technology. The main disadvantage of electrostatic actuators is the pull-in voltage limiting the linear displacement and the relatively high driving voltage.

Electromagnetic actuation offers a large scan angle, low driving voltage, and relatively more linear response than the competing actuation mechanisms. Despite the advantages of MEMS electromagnetic scanners, the achievable performance is limited by the large thermal dissipation inside a coil. In addition, magnets strong enough for high performance present significant volume and might require magnetic shielding. This leads to total package sizes that are larger than the competing MEMS actuators. Finally, they are complex and difficult to fabricate, particularly at small scales.

Electrothermal actuators present large scan angles at low driving voltages. They offer the largest fill factor compared to other categories of MEMS mirrors. However, the thermal response is relatively slow, but sufficient to perform real-time imaging. Since electrothermal actuators have relatively simple geometries, they are easy to fabricate and

can be made with a high fill factor. Electrothermal scanners seem to be excellent candidates to satisfy the requirements of OCT endoscopic imaging applications, however significant engineering effort is still needed to limit optical aberrations and to control scanner stability. Electrothermal scanners on silicon mirror plates that are suspended by a pair of torsion bars often present a temperature dependence in their oscillation behavior. This results in mechanical buckling of the beam. In this case, the initial position of mirror deflection can be difficult to control because the center of gravity of the oscillation system can be displaced from the initial position, changing the resonant mode and degrading the amplitude of oscillation when the scanner is driven at a constant frequency. In this case, the control of initial tilt angles can be difficult, causing optical alignment problems. The elastic constant of silicon can be a source of decreasing the temperature rise, generating a drift of resonance frequency—this is the case for all actuation mechanisms.

For all three actuation mechanisms, the one considered to have the most appropriate architecture seems to be the resonant scanner with gimbaled orthogonal single-axis mirrors. Here, the design needs to strike a balance where the combination of scan angle, resonance frequency, and mirror size is enough for definition of the desired resolution, while keeping the mirror optically flat to avoid image distortions. One important challenge with gimbaled dual-axis scanners is the possibility of crosstalk between the two axes. Another challenge for MEMS scanners developed for endoscopic applications is compact package size, requiring a package of around 10 mm$^3$ or less.

For all three mechanisms of actuation, a scanning mirror fails to maintain its flatness when it is mechanically oscillated at resonance frequency. The mirror surface should be sufficiently flat if the mirror curvature is less than 1 m, so as to not distort the beam. Mirror deformation leads to unwanted expansion of the reflected beam spot, degrading the image quality by blurring near the left- and right–hand side edges of the projected image. To overcome this problem, Hsu et al. [72] proposed to modify the mirror plate by generating a backside island to improve the rigidity of the mirror. Thus, masses far from the rotation axis are removed, keeping the resonant frequency high, while removing those in the central part. The structure of a gimbaled 2D scanner can be subjected to external vibration as well as intrinsic oscillations of the orthogonal axis. In particular, a scanner spinning about its *y*-axis at resonance frequency can be forced to tilt in the orthogonal *x*-axis by the gimbal structure because Coriolis forces are generated [73]. This last effect results in a small coupling of both axes, blurring the image near the edges. The solution is to design the scanner mechanism to have no resonant coupling, making the resonance frequencies of the orthogonal axes distant. It is also critical that the microscanner exhibits excellent accuracy of scan repeatability. The angular accuracy must be better than 1% of the angle step size.

One limitation of OCT probes using MEMS scanning mirrors arises because it is necessary to use a sinusoidal scan, as described in Section 2. The sinusoidal scan is less linear compared to linear scanning because the scan speed is not uniform at the center and at the edges of the B-scan because of non-uniformly spaced A-scans. The linearization of the scan requires driving the MEMS with higher harmonics of the scan frequency where parasitic vibrations of the resonance frequency appear since the MEMS scanning mirror does not have a closed loop feedback system. In this case, it is pertinent to implement a closed feedback loop, based on the integration of a piezoresistive strain gauge on the torsional beams of the scanning mirror to sense the beam strain and adjust the microscanner angle.

As discussed here, challenges need to be overcome to enable the implementation of MEMS scanners into endoscopic OCT systems, requiring minimally invasive recording of cross-sectional images in vivo with high resolution and high speed. To date, the electrothermal two-axis and resonant MEMS scanners seem to be the closest candidates to satisfy the requirements of endoscopic imaging applications under conditions to better control scanner stability. In conclusion, we hope to clearly demonstrate that MEMS scanner-based OCT probes offer several significant advantages and we expect them to have a bright future in mobile medical imaging devices.

**Table 1.** Comparison of three main actuation mechanisms used for scanning in OCT probes.

| | Mirror Size (mm)/Data from [46] | Angular Deflection (°)/Data from [46] | Resonance Frequency (Hz)/Data from [46] | Advantages | Drawbacks |
|---|---|---|---|---|---|
| **Electrostatic all** | | | | Fast response Low power consumption Large scan angle | Pull-in effect High driving voltage (50–100 V) Low force |
| Linear comb | ~1 | <10 Ave scan: 8.5 Ave piston: 29 μm | 250–5000 (ave 1800) | | |
| Vertical comb | ~1 | >10 Ave scan: 14 Ave piston: 94 μm | 150–10,000 (ave 5000) | | |
| *Characteristics* | electrostatic attractive force between conductors | a pair of electrodes with air gap | requires mechanical resonance to enhance scan angle | Small temperature dependance | Silicon DRIE and metallization |
| **Electromagnetic** | 0.8–3.5 | Maximum: 20 Ave scan: 15 Ave piston: 5 μm | 100–1000 | Larger driving Lower driving voltage Large scan angle | High power consumption External magnets increasing the size Electromagnetic interferences |
| *Characteristics* | Lorentz force between current and magnetic field | coil and permanent magnet | size limited by magnet | Multiple layers of metal and insulator for coil | Small temperature dependance |
| **Electrothermal** | 0.5–1 | ~25, maximum 64 Ave scan: 27° Ave piston: 270 μm | 117–1000 | Large scan angle Low driving voltage High fill-factor | High power consumption Slow response |
| *Characteristics* | thermal expansion by Joule effect | materials with different thermal expansion coefficients | multiple layers of metal and insulator for heater | Requires relatively high power | Bimorph sensitive to temperature change |

**Funding:** This work was supported by the collaborative project VIAMOS of the European Commission (FP7, ICT program, grant no. 318542), the ANR Labex Action program (ANR-11-LABX-0001-01), and received a support from Collégium SMYLE. It was also supported by the French RENATECH network and its FEMTO-ST technological facility. Work is supported by the Foundation for Polish Science co-financed by the EU under the European Regional Fund (project no MAB/2019/12).

**Conflicts of Interest:** The authors declare no conflict of interest.

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
