# Peer review of "MEMS Scanning Mirrors for Optical Coherence Tomography"

_photonics, doi:10.3390/photonics8010006_

Round 1

Reviewer 1 Report

The paper presented a good review about MEMS scanners for OCT applications but needs a considerable enhancement

The title is not appropriate since it gives the impression of reviewing the MEMS technologies for OCT while the paper focuses only on scanning mirrors. For example tunable filters and swept sources are not mentioned at all

English needs to be revised thoroughly with some highlighted examples in the attached version

Missing important class of integrated micro scanners based on translating a curved mirror using electrostatic actuator

Mentioned only the architecture of Michelson interferometer and low coherence light. Missing important architecture of swept laser source, where the latter has been demonstrated using MEMS technology

Quality of figures overall needs improvement

Author Response

Response in a word file

Reviewer 2 Report

The manuscript contains a good description of each actuator group working principle and moreover, points out the important characteristics of the actuator group. It describes the advantages and disadvantages of each actuator type in regards to OCT application and it is well illustrated in Chapter 4. A nice overview of all actuation principles is given in Table 1.

However, the manuscript in the present form is confusing and misleading as the title and the scope of the manuscript do not match. The title of the manuscript is “MEMS technologies for OCT” while the whole manuscript focuses on MEMS mirrors for endoscopic OCT. The structure of the manuscript is also a bit chaotic and needs improvement. In the comments below, authors can find points of improvement for this manuscript to be publishable.

Comment 1:

Authors have to decide if they want to focus the scope of the manuscript towards endoscopic OCT or OCT in general. However, in the present form, I do not see a big difference in review contribution for endoscopic OCT MEMS mirrors compared to J.Sun and H. Xie “MEMS-Based Endoscopic Optical Coherence Tomography”, https://doi.org/10.1155/2011/825629, or more recent review paper from Z. Qiu and W. Piyawattanametha “MEMS Actuators for Optical Microendoscopy”, https://doi.org/10.3390/mi10020085. I believe the scientific community would benefit more if the manuscript is according to the title, where all MEMS technologies and applications are covered not only endoscopic OCT with MEMS mirrors focusing on

Comment 2:

The title is “MEMS technologies for OCT” but the whole manuscript is referring to MEMS micromirrors. This means the manuscript implies the scanning is only possible using micromirrors, which is not true. The optical scanning can be achieved in different ways still using MEMS technologies, for example using fiber scanning and it is demonstrated in Y.H. Seo et al, “Electrothermal MEMS fiber scanner for optical endomicroscopy” https://doi.org/10.1364/OE.24.003903, in X. Zhang et al, “A non-resonant fiber scanner based on an electrothermally-actuated MEMS stage”, http://dx.doi.org/10.1016/j.sna.2015.07.001 and in A. Acemoglu et al, “Design and Control of a Magnetic Laser Scanner for Endoscopic Microsurgeries”, https://doi.org/10.1109/TMECH.2019.2896248.

Also, optical scanning can be achieved by lens movements as well and it is demonstrated in L. Zhou et al., “A MEMS lens scanner based on serpentine electrothermal bimorph actuators for large axial tuning”, https://doi.org/10.1364/OE.400363, or in A. Jovic et al. “A highly miniturized single-chip MOEMS scanner for all-in-one imaging solution”, https://doi.org/10.1109/MEMSYS.2018.8346472.

Although the mentioned papers do not demonstrate OCT image itself, for sure they represent MEMS technology which can be used in OCT. All mentioned examples have smaller assembly complexity than MEMS mirror-based systems. Moreover, currently, there is a lot of development of MEMS scanners for LIDAR applications and a lot of them are applicable for OCT as well.

Comment 3:

There are only 12 references out of 69 which date back from 2016 and onwards out of which 2 are review papers and 6 are authors work. For sure there are more groups and published papers in the last 5 years with different MEMS technology working on optical scanners for OCT or at least applicable to OCT. Out of 5 examples of papers given in Comment 2, 3 are from 2018 and onwards. Please update the review manuscript with recent references.

Comment 4:

Replace MEMS mirrors/micromirrors with MEMS scanners/microscanners when talking about general optical scanning.

Comment 5:

Rephrase text in lines 69-87 and adjust Eq 1 to match general requirements for OCT optical scanners, not only mirror-based. Also, Change Fig 2 and Fig 3 to match general MEMS scanners, not only mirror-based.

Comment 6:

Lines 141-162 refer only to MEMS mirrors. Move to a chapter that explains mirror-based systems, i.e. Chapter 4.

Comment 7:

Change Chapter 3 to cover requirements for different OCT applications, not only endoscopes. There are handheld devices for both ophthalmology and dermatology as well which use MEMS can benefit a lot from MEMS technology.

Comment 8:

Chapters 4 and 5 have the same name.

Comment 9:

Separate each actuator type group as a separate chapter. The theoretical background from Chapter 4 can be used as an intro to each group.

Comment 10:

The same attention is given to 2 papers described in 5.2 as to 7 papers given in 5.1. Is this work more significant than all different designs of 5.1?

Comment 11:

I personally really like the concept from 5.2. It reminds me of the big telescopes used in astronomy but miniaturized. However, although miniaturized, the device footprint is 15x15 mm, and by design, it should be in a forwarding view probe. For such a design, it would require a probe diameter of 22mm which is quite thick for an endoscope? I don't see this fitting in an endoscope, at least I would not line such an endoscope in my arteries...

In the text, it is also mentioned the device is for the dermatology application. Once again the question is: Is the focus on endoscopes for MEMS technology in general? Please make a decision about the scope of the manuscript and make a choice of a selected state of the art accordingly.

Comment 12:

Are there no other magnetically driven OCT scanners than one presented in [59]?

Comment 13

Why 5.4 and 5.5 are so different they have to be in separate chapters? The working principle of both mirrors are the same, and 5.5 described work is a buildup of the mirror given in 5.4. Also, 3 more electrothermally actuated MEMS scanner concepts are previously mentioned in Comment 2 which are very different from MEMS mirrors. The "MEMS technology for OCT" should cover more than just mirrors.

Comment 14

The conclusion must be adjusted to match the revised manuscript which should cover a broad range of MEMS technologies, not only focusing on mirrors in endoscopes.

Author Response

Response in a word file

Round 2

Reviewer 1 Report

The authors have made significant improvement in the revised version. But it still missing important work about MEMS technologies for Optical Coherence Tomography based on in-plane configuration for light propagation parallel to the plane of the substrate. This configuration allows the integration of different sub-systems on-chip I recommend the authors include and cite the work related to:

  1. Deeply-Etched Filter for Swept Laser Sources using Electrostatic Parallel Plate Actuator
  2. Deeply-Etched Wide-Angle Scanner using Electrostatic Comb-Drive Actuator
  3. Similar work

Author Response

These references are not appropriate for CT - I cannot include, sorry

Reviewer 2 Report

Dear author,

The new version of the manuscript is very good and has a nice reading flow. The change of title and presented text are now aligned. With additional references and moreover a nice explanation of actuation principles this manuscript is distinguishable compared to other review papers in this field. I understand that it was impossible to cover all technologies within such a short period. However, I still have two comments and you can find them below.

Comment A:

We have a minor disagreement regarding your statement in Comment 11 "Often, a miniature imaging system is considered as an endoscopic application.". Endoscopes are tubular instruments used to assess deep into very narrow channels. In medical applications used to access blood vessels, digestion organs and etc. Therefore, in the whole manuscript (especially in section 3), the "endoscopic OCT" should be replaced with "OCT probe" from the whole section whenever it does not specifically describe an endoscopic application. Especially as the work presented in Fig 11 is not for an endoscope.

Comment B:

In addition, work in Fig 11 and Fig20 significantly simplifies probe assembly and packaging so it should be stressed out.

Author Response

Ok I changed